# Can Acute Neurological Disease Cause Cardiomyopathy in Horses?

**DOI:** 10.3390/ani15101447

**Published:** 2025-05-16

**Authors:** Valentina Vitale, Ana Velloso Álvarez, María de la Cuesta-Torrado, Patricia Neira-Egea, Marie Vandecandelaere, Elizabeth Tee, Marina Gimeno, Gaby van Galen

**Affiliations:** 1Hospital Clínico Veterinario, Departamento de Medicina y Cirugía Animal, Universidad Cardenal Herrera-CEU, CEU Universities, Alfara del Patriarca, 46115 Valencia, Spainmaria.de2@uchceu.es (M.d.l.C.-T.);; 2Sydney School of Veterinary Science, Faculty of Science, The University of Sydney, Sydney, NSW 2006, Australiagaby@equinespecialists.eu (G.v.G.); 3Goulburn Valley Equine Hospital, Congupna, VIC 3633, Australia; 4Department of Primary Industries, Elizabeth Macarthur Agricultural Institute, Woodbridge Road, Menangle, Sydney, NSW 2568, Australia

**Keywords:** equine, cardiac disease, myocardial necrosis, catecholamine, seizures

## Abstract

In human medicine, neurological diseases have been associated with transient cardiac abnormalities. Rodent and canine models of cardiac injury after brain trauma have been developed; however, this condition has not previously been reported in horses. In this case report, we describe three horses with a diagnosis of acquired cardiomyopathy following acute neurological signs due to different causes, thereby raising the question of whether myocardial injury in horses can be induced by neurological disease similar to that in humans. Although we cannot prove this theory, based on the human literature, a similar pathophysiology can also occur in the equine species. We suggest that horses with acute neurological diseases benefit from cardiac monitoring to identify the presence and severity of myocardial damage and apply further treatment if needed.

## 1. Introduction

Myocardial disease is rarely diagnosed in horses. It can be associated with a wide variety of causes, most commonly infectious and toxic aetiologies [1,2].

In human medicine, neurological diseases have been associated with transient cardiac abnormalities [3,4]. Rodent and canine models of cerebral cardiac injury after brain trauma have been developed [5,6]; however, this condition has not previously been reported in horses.

Neurological injuries in humans can cause myocardial alterations, which may manifest as reversible neurogenic left ventricular dysfunction, known as neurogenic stunned myocardium (NSM). Alternatively, these injuries may result in more severe conditions such as myocardial ischemia and infarction (MI) [7,8]. NSM is a form of stress cardiomyopathy in which different acute neurological pathologies, such as intracerebral haemorrhage, traumatic brain injury, acute ischemic stroke, seizures, encephalitis, and brain tumours, among others, could trigger acute myocardial dysfunction [9]. There is still controversy regarding the underlying pathophysiological mechanism, but recent evidence highlights the role of autonomic dysfunction [6]. In particular, catecholamine-induced myocardial alterations are supported by the fact that disrupting the sympathetic chain at the cervical level halts the arrhythmias, while vagotomy does not [3]. Cardiac troponin levels rise significantly more in myocardial infarction (MI) compared to neurogenic stunned myocardium (NSM). A cut-off value has been established to distinguish between these two conditions, which helps in assessing anaesthetic risks and prognosis [3].

The aim of this case report is to describe three horses with a diagnosis of acquired cardiomyopathy following acute neurological signs due to different causes, thereby raising the question of whether myocardial injury in horses can be induced by neurological diseases similar to those in humans.

## 2. Case Summary

### 2.1. Case 1

A 5-year-old 80 kg Shetland pony stallion was referred to the Camden Equine Hospital of the University of Sydney due to depression, anorexia, and fever for 1 week following an episode of seizures caused by an accidental intra-arterial injection of xylazine (unknown dose). Before the seizures, the stallion was reported to be healthy, and the owners had not noted any clinical signs that could have indicated pre-existing cardiac problems. The sedation was a premedication to perform standing castration in the field, a procedure that was not performed due to the complication that occurred. During the days prior to admission, the referring veterinarian started a treatment with flunixin meglumine and penicillin procaine (details of these and the following treatments can be found in Table 1), but no improvement was observed; thus, the pony was referred for further assessment.

On admission, the pony showed marked tachycardia (100 bpm) and tachypnoea (120 rpm). On thoracic auscultation, muffled heart sounds but no abnormal respiratory sounds were detected.

Abdominal and thoracic ultrasound (US) examinations (Philips EPIQ 5, Koninklijke Philips, Eindhoven, The Netherlands) were unremarkable. On echocardiography, moderate pericardial effusion (2 cm of anechoic fluid) was observed. Immediately, intranasal oxygen administration was started and a pericardiocentesis from the left fourth intercostal space was performed as previously described [1]. A mild turbid orange fluid was obtained with a nucleated cell count of 0.5 × 103/µL (reference range < 1.5 × 103/µL) with 13% neutrophils, total protein (TP) of 4 g/dL (reference range < 2.5 g/dL), and lactate of 2.3 mmol/L (reference range < 2 mmol/L), compatible with a modified transudate.

Subsequently, the respiratory rate (RR) decreased to 40 rpm while the heart rate (HR) remained elevated (100–150 bpm). On repeated echocardiography, from the right parasternal 4-chambers view, the myocardium appeared thickened with an area of decreased echogenicity on the outside layer, especially on the left ventricular free wall (LVFW). The cardiac chambers appeared normal in size and no regurgitations were observed from any of the valves (Figure 1, Appendix A). Measurements obtained in M-mode included interventricular septum in systole (IVSs) and diastole (IVSd), left ventricular internal diameter in systole (LVIDs) and diastole (LVIDd), and left ventricular free wall in systole (LVFWs) and diastole (LVFWd). Furthermore, the left ventricular (LV) fractional shortening (FS, in %) was calculated as FS = (LVIDd—LVIDs)/LVIDd × 100; and relative LV wall thickness at end-diastole (RWT) was calculated as RWT = (IVSd + LVFWd)/LVIDd. Detailed M-mode measurements are reported in Table 2, while haematological results can be found in Table 3. The biochemistry was unremarkable. Serum cTnI was markedly increased at >25 ng/mL (<0.02 ng/mL). A diagnosis of cardiomyopathy was reached and treatment with intravenous fluid therapy, penicillin, gentamicin, flunixin meglumine, and dexamethasone was started. Furthermore, an ECG Holter device (Televet, Jorgen Kruuse, Langeskov, Denmark) was placed for continuous monitoring. During the following 12 h, frequent ventricular premature complexes (VPCs) progressing to monomorphic ventricular tachycardia (VT) with an HR of 130–150 bpm were observed. A continuous rate infusion (CRI) with lidocaine was started. Since the VT was non-responsive, magnesium sulphate CRI was commenced. Finally, 48 h after admission and treatment, the pony converted to sinus rhythm with an HR of 70 bpm. Gradually, IV fluid therapy and the CRI’s of magnesium sulphate and lidocaine were discontinued, as was antimicrobial therapy. Dexamethasone was changed to oral prednisolone. The pony was discharged 8 days after admission with an HR of 68 bpm and RR of 16 rpm. The recommendations were to maintain the pony on the current treatment for 10 days and return for a control examination with echocardiography, ECG, and serum cTnI measurements. Due to economic constraints, the pony did not return and was euthanized by the referring veterinarian approximately two weeks after discharge due to a suspected recurrence of VT. No post-mortem examination was performed.

### 2.2. Case 2

A 20-year-old crossbred mare was referred to the Hospital Clínico Veterinario of the Universidad CEU-Cardenal Herrera due to an open fracture of the left maxillary bone with copious epistaxis from both nostrils. The causes and circumstances of the head trauma were unknown. The haemorrhage was controlled by the referring veterinarian by gauze packing the fracture site and maxillary sinus. Phenylbutazone was administered before referral (details of this and the following treatments can be found in Table 1). The owners reported that before the trauma the mare was healthy and did not exhibit any clinical signs suggesting pre-existing cardiac disease.

On admission, an open wound approximately 3 cm in diameter above the left maxillary bone was observed. The mare presented tachycardia (80 bpm), pink dry mucous membrane, normal respiratory rate (20 rpm), and rectal temperature (37.4 °C). Head radiographs showed a comminute fracture of the left maxillary and zygomatic bones, and fragmentation areas of the facial crest. Severe inflammation of the left nasal meatus with several blood clots was observed on upper airway endoscopy. The haematological results can be found in Table 3, while the biochemistry was unremarkable.

After sedation with detomidine and butorphanol, the wound was debrided, and small fragments were extracted from the fracture site and the left nasal concha. A foley catheter was placed within the fracture and sutured to the skin, and a bandage was applied to cover the wound. The following treatment was started: IV fluid therapy, cefquinome, phenylbutazone, and dexamethasone. The choice of the antimicrobial was based on the preference for an injectable antimicrobial that could cross the blood–brain barrier. Trimethoprim–sulfadiazine was considered, but due to the previous administration of detomidine, was not elected to avoid the development of arrhythmias [11]. During the following 12 h, the HR decreased to 40 bpm but the mare showed persistent mild, bilateral epistaxis. Tranexamic acid and etamsylate were administered as a single-dose therapy.

Approximately 24 h after admission, a disconnection of the extension set of the IV catheter was recorded through video camera surveillance. A veterinarian entered the stable and reconnected the fluid set within approximately 30 s of the event; however, the mare immediately developed severe respiratory signs with tachypnoea (60 rpm) and recumbency. Within 30 min, neurological signs with diffuse fasciculations and hyperesthesia of the hindlimbs appeared, which rapidly progressed to focal and secondary generalised seizures, loss of consciousness, and tonic–clonic contractions of the limbs. The emergency treatment consisted of the administration of intranasal humidified oxygen and boluses of IV diazepam. However, as the seizures recurred on three occasions within a few hours, phenobarbital was administered at tapered doses for three days. Furthermore, dimethyl sulfoxide and 7.5% hypertonic saline were added to prevent brain oedema. Three more episodes of partial seizures of lower magnitude with the fasciculation and twitching of head and neck muscles, together with spontaneous horizontal nystagmus, were recorded during the following 12 h. The mare developed transient blindness that spontaneously resolved within the next 24 h and no other neurological signs were observed thereafter. On the 3rd day after admission, the mare developed progressively increasing tachycardia (up to 80 bpm) and marked anaemia (PCV 10%). Ongoing sinus haemorrhage was suspected; thus, five litres of fresh blood were administered, followed by 3 additional litres of fresh-frozen plasma to replace blood loss and support oncotic pressure.

On ECG (Televet, Langeskov, Denmark), frequent VPCs and short periods of self-limiting VT were detected. Ecocardiography (LogiqTM P10XDclearTM, General Electric Company, Cincinnati, OH, USA) was performed and the same parameters as in Case 1 were recorded. On subjective assessment, a focal hyperechogenic area of approximately 2 cm was observed within the basal portion of the IVS, surrounded by a larger hypoechogenic area, observed in multiple different planes (Figure 2, Appendix A). Basal and mid-ventricular dyskinesia of the IVS was also observed. Furthermore, mild aortic and tricuspid regurgitation were detected. Detailed M-mode measurements are reported in Table 2. Serum cTnI was markedly increased at >25 ng/mL (<0.02 ng/mL). Based on these findings, cardiomyopathy was diagnosed, and lidocaine CRI was added to the treatment. Additionally, a persistently elevated mean arterial pressure (160 mmHg) was recorded by non-invasive blood pressure monitoring (BU A57, Medisana GmbH, Neuss, Germany) at the tail level. Concomitant hypernatremia of 150–160 mmol/L (135–145 mmol/L) without other electrolyte disturbances was also observed. At this stage, her pain score was estimated between 4 and 6 according to the Equine Utrecht University Scale for Facial Assessment of Pain (EQUUS-FAP), suggestive of a low level of pain [12]. Hypertonic saline administration was discontinued, while the following drugs were added to the treatment: acepromazine as a hypotensive, gabapentin as an anti-sympathetic, furosemide to stimulate renal sodium excretion, and a synthetic colloid to support oncotic pressure. Despite the treatment, during the 5th–7th days of hospitalisation, mild pulmonary and severe ventral and peripheral limb oedema developed. However, tachycardia, hypertension, hypernatremia, oedemas, and anaemia progressively resolved. Treatments were gradually discontinued, and antimicrobial therapy was switched to oral trimethoprim–sulfadiazine.

The mare was discharged 23 days after admission, with an HR of 40 bpm, a PCV of 25% (30–45%), a serum total protein concentration of 5.6 g/dL (5.5–7.5 g/dL), and a serum cTnI concentration of 0.06 ng/mL (<0.02 ng/mL). Recommendations were made to maintain the antimicrobial treatment for one more week and return for a recheck, including an upper airway endoscopy, echocardiography, and serum cTnI measurements. However, due to economic constraints, the mare never returned; the referring veterinarian reported that the mare is well and maintains normal general examinations.

### 2.3. Case 3

A 4-year-old thoroughbred gelding was referred to the pathology service of the University of Sydney due to sudden death following a backward fall. The racehorse was previously healthy and in training. Immediately after the fall, the gelding showed generalised seizures and rigidity of both hindlimbs and the right foreleg. By the time the referring veterinarian examined the animal, it was in left lateral recumbency with a large amount of blood and oedema originating from both nostrils, fixed and dilated pupils with no pupillary reflexes, and no signs of voluntary movement. The signs were reported as rapidly progressive, and the horse died shortly after the arrival of the veterinarian.

A post-mortem examination was conducted 7 h after death. The findings were compatible with traumatic brain injury caused by blunt force trauma. There were focally extensive comminute fractures in the right rostral maxilla, zygomatic, and lacrimal bones. The brain was severely oedematous and congested with rostral herniation of the cerebellum. Generalised pulmonary congestion and oedema were diagnosed. Within the endocardium, epicardium, and myocardium multifocal to coalescing areas of dark red discolouration were seen, compatible with haemorrhages. Moreover, diffuse severe congestion and haemorrhages were found in both kidneys.

Histopathology examination of the brain confirmed diffuse moderate spongiosis, mild gliosis, multifocal neuronal necrosis, and perivascular and meningeal oedema within the brain. In the brainstem, axonal degeneration and moderate spongiosis were found. Furthermore, affecting 20–30% of the myocardium, there were multifocal to coalescing areas of acute haemorrhages and coagulative necrosis (Figure 3).

## 3. Discussion

This report presented three horses that developed cardiomyopathy following the acute onset of seizures as a consequence of intraarterial injection, venous air embolism, or head trauma.

In human medicine, acquired cardiomyopathy associated with neurological disease is a well-described pathology and it has been related to many types of neurological events [3,6,13,14,15]. The pathophysiological mechanism of this type of myocardial injury is commonly attributed to catecholamine excess, but systemic inflammation and neuroendocrine derangements may also contribute to its development [8]. Differentiating between reversible NSM and MI is critical in human medicine to establish the risks related to early surgical interventions. A cut-off of 2.8 ng/mL for cardiac troponin has been established, indicating a lower risk of cardiovascular deterioration for patients with NSM [3]. Furthermore, the 12-lead ECG is an important diagnostic tool for the detection of MI as it is characterised by ST-elevation [16]. In a horse with experimentally induced MI, ST-segment changes were detected using a 12-lead ECG, resembling those observed in humans [17]. However, in horses, ECG recordings are generally used to detect arrhythmias and often only the base-apex derivation is employed [18]. As in the cases described here, where the holter device was applied using the base-apex derivation, it is possible that subtle changes in the ST-segment may not have been observed.

Cardiomyopathy associated with neurological disease has not been previously reported in horses, but based on the cases presented here, a similar pathophysiological mechanism can be suspected.

Cases 1 and 3 were young animals and, although no cardiac examination was performed prior to the neurologic event, they were reported to be healthy. In Case 2, the blood loss, the trauma, and the venous air embolism could also have contributed to the development of cardiomyopathy [19,20].

Furthermore, the hypertension observed in Case 2 could be attributed to pain or excessive fluid, but neither of these could be identified. Alternatively, the hypertension could be related to the neurological disease. Traumatic brain injury, cerebral hypoxia, and cerebrovascular accidents in humans can lead to a condition known as paroxysmal sympathetic hyperreactivity, which is characterised by hypertension and tachycardia, among other symptoms related to autonomic dysregulation and a surge of catecholamines [21,22]. Neurological signs in this case were attributed to the venous air embolism due to the temporal relationship; however, the mare experienced a head trauma 24 h earlier; thus, neurological signs could also be a consequence of the initial trauma.

The concomitant pulmonary oedema developed by horses 2 and 3 may be associated with pulmonary microvascular injury, as well as with ventricular dysfunction secondary to myocarditis, similar to those reported in people with NSM [13]. The potential development of hypertension and pulmonary oedema highlights the importance of the close monitoring of IV fluid therapy in patients with NSM.

While horse 3 rapidly died due to the severe brain injury, horses 1 and 2 recovered. In both horses, the increase in cTnI was so high that it exceeded the laboratory upper limit of detection. In humans, an initial cTnI level 10-fold higher than the upper limit of the reference range is associated with MI [3] and poor in-hospital outcome [23], and these two horses had exceeded this level. In equine patients, although cTnI is routinely used in clinical practice [24], there is a lack of reference values to distinguish between reversible and irreversible myocardial injury. Additionally, given the difference in myocardial mass between horses and humans, cTnI levels are likely not directly comparable in these two species.

Based on the echocardiographic findings of the two cases, horse 1 had a larger estimated affected area with myocardial damage (both apex and LVFW) compared to horse 2, which only had a focal area within the IVS. Both cases presented an increased thickness of the IVS and LVFW both in systole and diastole, reduced LVIDs, and significantly increased FS. However, the increase in FS was higher in horse 1, which also presented a reduction in LVIDd, suggesting a more severe LV dysfunction. Furthermore, horse 1 was presented to the hospital 1 week after the neurological event, while horse 2 was already hospitalised when the neurological signs occurred, leading to an earlier diagnosis and treatment. In laboratory animals, immediate hypokinetic cardiac effects are followed by chronic changes typically leading to cardiac hypertrophy and remodelling with increased collagen in the LV wall [6,25]. The increased thickness of the IVS and LVFW observed in cases 1 and 2 could be compatible with this form of hypertrophic cardiomyopathy; however, the decreased echogenicity could also suggest the presence of haemorrhages and coagulative necrosis within the myocardium as observed in Case 3. Moreover, the decrease in LVIDs associated with the increase in FS are changes commonly observed after exercise or stress echocardiography in horses [26,27,28] and are attributed to sympathetic stimulation [29]. Thus, it is possible that our patients, despite the extremely high value of cTnI, were only showing a cardiac response to the increased catecholamine levels and experienced less hypokinetic LV dysfunction compared to what has been observed in human patients.

Due to the delay in which horse 1 was referred, the cardiomyopathy could have been in a more advanced stage, which could explain the more severe LV dysfunction with higher FS and refractory VT.

The short- and long-term prognosis of NSM in humans was at first considered fair to good as the condition was thought to be mostly benign [30]. However, between 20 and 52% of in-hospital follow-up of stress cardiomyopathy patients revealed various cardiac complications such as [31] LV systolic dysfunction to acute heart failure, cardiogenic shock, arrhythmias, and intraventricular thrombosis that can change the course of the disease, and even lead to death [13,23]. Most human patients with NSM exhibit mild symptoms and typically only require cardiac monitoring, with no specific therapy necessary [8,23]. Many equine neurological cases might be overlooked for the presence of cardiomyopathy, especially those with mild lesions. This is because analysis such as cTnI concentrations, ECG, and echocardiography are typically not performed unless overt clinical cardiac signs are present.

The limitations of this report are the absence of cardiac examinations of the cases prior to the onset of clinical disease, the lack of follow-up examinations on horses 1 and 2, and the absence of a post-mortem examination of horse 1. Furthermore, we did not perform two-dimensional speckle-tracking echocardiography nor myocardial biopsy to confirm our hypothesis, but myocardial histopathology was performed on Case 3. With the three described cases, no clear causative link could be established between the neurological signs and the cardiomyopathy. Despite these limitations, the temporal relationship between the neurological events and the development of cardiomyopathy, along with the similarities to the human literature, strongly suggests that the two events were related in the three cases described here.

## 4. Conclusions

This report describes three cases with myocardial injury following neurological disease. Although this observation does not deliver proof for a causal relationship, based on the available human literature on the topic, these cases suggest that it is likely and plausible that cardiomyopathy of neurologic origin can also occur in horses following acute brain injury of different causes. Based on this report, we suggest that horses with acute neurological diseases benefit from cardiac monitoring, including continuous ECG and plasma cTnI measurements, to identify the presence and severity of myocardial damage and apply further treatment if needed.

## Figures and Tables

**Figure 1 animals-15-01447-f001:**
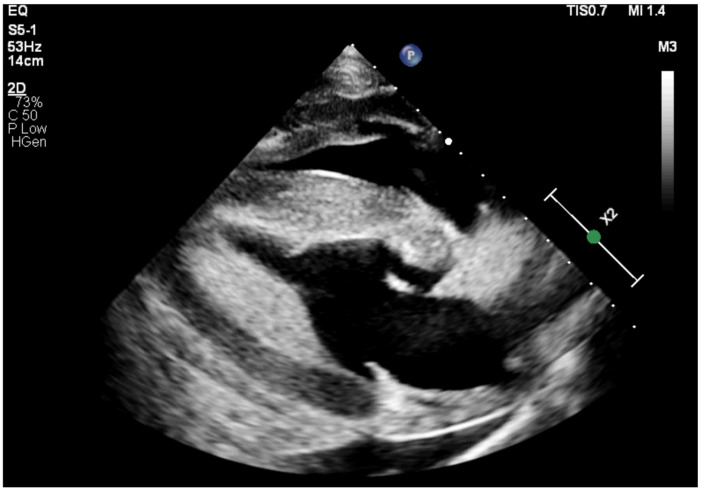
Right parasternal long-axis 4-chambers view of horse 1: subjective increased thickness of interventricular septum (IVS) and left ventricle free wall (LVFW) and decreased echogenicity of the outer layer of the apex and LVFW.

**Figure 2 animals-15-01447-f002:**
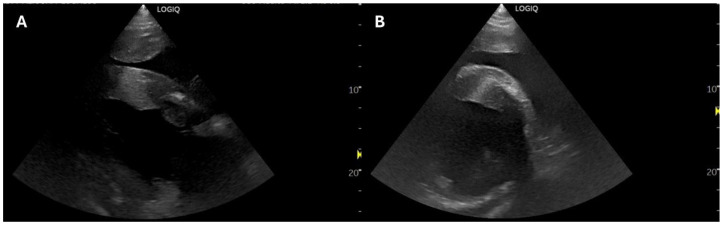
The right parasternal long-axis 4-chambers view of horse 2: the subjective increased thickness of interventricular septum (IVS) and the right ventricle free wall and focal hyperechogenic area within the IVS rounded by a wider hypoechogenic area (**A**). The right parasternal short-axis view just above the papillary muscle level of horse 2: the same focal hyperechogenic area within the IVS rounded by a wider hypoechogenic area (**B**).

**Figure 3 animals-15-01447-f003:**
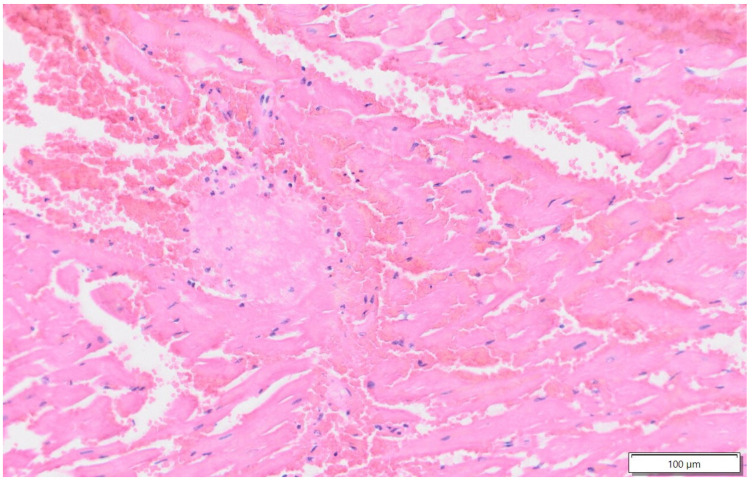
Myocardium, Case 3, H&E. Separating the muscle fibres, there are areas of extravasated free red cells (haemorrhage). The cardiomyocytes are hypereosinophilic and hyalinized with fragmented myofibrils, a loss of cross-striations, and nuclear pyknosis or karyolysis (necrosis).

**Table 1 animals-15-01447-t001:** Details of the treatments administered to cases 1 and 2.

Drugs	Case	Dose	Reference
Flunixin meglumine	1	1.1 mg/kg q 24 h, IV	Ilium Flunixil, Glendenning, NSW, Australia
Penicillin procaine	1	22,000 IU/kg q 12 h, IM	Ilium Propercillin, Glendenning, NSW, Australia
Hartmann’s intravenous fluids	1	2 mL/kg/h	Baxter Healthcare, Toogabbie, NSW, Australia
Gentamicin	1	6.6 mg/kg q 24 h, IV	Gentam-100, Glendenning, NSW, Australia
Dexamethasone	1	0.2 mg/kg q 24 h, IV	Ilium Dexapent, Glendenning, NSW, Australia
Lidocaine	1	loading dose of 0.3 mg/kg IV over 15 min, followed by 0.025 mg/kg/min	Lignocaine 20, Glendenning, NSW, Australia
Magnesium sulphate	1	2 mg/kg/min IV	Pfizer Australia Pty Ltd., Sydney, NSW, Australia
Prednisolone	1	1 mg/kg q 24 h, PO	Pred-X 20 mg, Dechra Veterinary products PTY LTD, Somersby, NSW, Australia
Phenylbutazone	2	4.4 mg/kg q 12 h, IV	Butasyl^®^, Zoetis Manufacturing & Research Spain, Girona, Spain
Detomidine	2	0.006 mg/kg IV	Domidine^®^, Eurovet Animal Health B.V., Bladel, The Netherlands
Butorphanol	2	0.03 mg/kg IV	Torbugesic^®^, Zoetis Manufacturing & Research Spain, Girona, Spain
Ringer Lactate	2	Initially a 10 L bolus, and subsequently at a rate of 2 mL/kg/h	B. Braun Medical SA, Rubí, Barcelona, Spain
Cefquinome	2	1 mg/kg q 24 h, IM	Cobactan 2.5%^®^, Intervet International GmbH, Unterschleißheim, Germany
Dexamethasone	2	0.1 mg/kg q 24 h, IV	Caliercortin 4 mg/mL^®^, Laboratorios Calier, S. A., Les Franqueses del Vallès, Barcelona, Spain
Tranexamic acid	2	10 mg/kg IV	Amchafibrin 500 mg^®^, Meda Pharma SL, Madrid, Spain
Etamsylate	2	5 mg/kg IV	Hemo 125 mg/mL^®^, Zoetis Manufacturing & Research Spain, Girona, Spain
Diazepam	2	0.5 mg/kg IV	Valium^®^, Atnahs Pharma Netherlands B.V., København S, Denmark
Phenobarbital	2	2 mg/kg q 12 h, IV	Luminal^®^, KERN PHARMA, Terrassa, Barcelona, Spain
Dimethyl sulfoxide	2	0.2 g/kg q 12 h, IV diluted 10% in physiological saline solution	Dimetil sulfoxide, Fagron Ibérica SAU, Terrasa, Barcelona, Spain
Hypertonic saline	2	5 mL/kg IV	B. Braun Medical SA, Rubí, Barcelona, Spain
Acepromazine	2	0.03 mg/kg q 8 h, IM	Equipromazina^®^, Labiana Life Sciences, S.A., Terrassa, Barcelona, Spain
Gabapentin	2	20 mg/kg q 12 h, PO	Gabapentina Sandoz^®^, Sandoz Farmacéutica, S.A., Madrid, Spain
Furosemide	2	1 mg/kg q 12 h, IV	Seguril^®^, Sanofi-aventis, S.A., Barcelona, Spain
Synthetic colloid	2	10 mL/kg q 24 h, IV	Gelaspan^®^, B. Braun, Melsungen, Germany
Trimethoprim-sulfadiazine	2	30 mg/kg q 12 h, PO	Equibactin^®^, Dechra Regulatory B.V., Bladel, The Netherlands

**Table 2 animals-15-01447-t002:** M-mode echocardiographic measurements of cases 1 and 2 compared with reference ranges for their weight (for horse 1, ranges were calculated using regression equations from the weight) [10].

Parameters	Horse 1	Range for 80 kg	Horse 2	Range for 454–620 kg
IVSd (cm)	1.83 *	1.32–1.68	3.24 *	2.6–3.0 cm
LVIDd (cm)	4.37 *	5.33–6.47	10.54	10.4–12 cm
LVFWd (cm)	1.75 *	1.02–1.38	3.18 *	2.2–2.8 cm
IVSs (cm)	2.86 *	2.08–2.52	5.33 *	4.1–5.1 cm
LVIDs (cm)	2.10 *	3.54–4.66	5.10 *	6.5–8.1 cm
LVFWs (cm)	2.59 *	1.6–2.2	4.2 *	3.5–4.1 cm
SF (%)	82 *	32.3–40.1	51.6% *	32.3–40.1%
RWT	0.82	-	0.61	0.41–0.63

IVS = interventricular septum; LVID = left ventricular internal diameter; LVFW = left ventricular free wall; SF = shortening fraction; RWT = relative wall thickness; d = diastole; s = systole. * Indicates altered result.

**Table 3 animals-15-01447-t003:** Haematology and cardiac troponin I (cTnI) results of horses 1 and 2 with the reference values.

Blood Parameter	Horse 1	Horse 2	Normal Range
RBC (×10^6^/μL)	5.94 *	5.52 *	6.5–12.5
Haemoglobin (g/dL)	9.5 *	9.6 *	11–19
PCV (%)	25 *	26 *	30–45
WBC (×10^3^/μL)	8.69	17.09 *	5.5–12
Neutrophils (×10^3^/μL)	5.21	15.67 *	2.5–8.1
Lymphocytes (×10^3^/μL)	2.43	0.77 *	1.63–3.40
Monocytes (×10^3^/μL)	1.04 *	0.53	0.0–0.7
Eosinophils (×10^3^/μL)	0.00	0.09	0.00–0.96
Basophils (×10^3^/μL)	0.00	0.02	0.00–0.36
TP (g/dL)	7	4.6 *	5.3–7.3
Fibrinogen (mg/dL)	500 *	200 mg/dL	<400
Platelets (×10^3^/μL)	124	92	80–300
Lactate (mmol/L)	3.2 *	6.9 *	<2
cTnI (ng/mL)	>25 *	>25 *	<0.02

RBC = red blood cells; PCV = packed cell volume; WBC = white blood cells; TP = total proteins; cTnI = cardiac troponin I. * Indicates altered result.

## Data Availability

Further images or video loops from the ultrasounds of Cases 1 and 2 are available upon request.

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
