# Peer review of "Can Acute Neurological Disease Cause Cardiomyopathy in Horses?"

_animals, 2025, doi:10.3390/ani15101447_

Round 1
Reviewer 1 Report
Comments and Suggestions for Authors
The authors have written a case report regarding the potential connection between neurological disease and myocardial lesion, and therefore, dysfunction.
Some minor comments below, and I will advise for publication.
Introduction
Line 50-51: first part of the phrase, before the comma, I think needs clarification with the “be”.
Case 1- Why did the authors decide to give AB, even though with normal WBC ?
Does the fever was related to the cardiac imbalance, or you believe could have had a bacterial involvement as well, since we had arterial punction?
Line 87: can you describe the technique for periocardiocentesis ? And in the text can you add a reference “as described..” ?
Case 2- One limitation is that no cardiac assessment has been made before, although, the clinical evidence is unquestionable (as above). Why the use of 7.5% hypertonic saline after the seizures? For potential brain edema?
Line 170: there is a “8” after the final point.
Line 192: "partial seizures”, can the authors please clarify and potentially point the difference between this episodes and the ones before?
Can the authors say what the window of access of video S3?
At the moment of the hypernatremia, was she still having seizures or neurological signs?
Line 216: there is “9” lost.
Figure 3: no inflammatory infiltrate or cells were seen in the necrosis areas? I think the legend of figure 3 is not correct.
Can the authors edit table 1 and table 2 with same size and layout?
I congratulate the authors for their work. The cases are very interesting and the pathological basis for the cardiac lesions is of deeper research ! One can speculate that the neurologic damage, affect the cardiac tissue. Further anatomopathological studies are needed in the site of the lesion, to verify the status of the innervation and cross results with the same analysis of the SNC area that send nerve branch’s to the heart. Potentially there is a neutrophilic inflammatory venue (giving the acute presentation). There is nerve fiber, SP positive, that are known to release inflammatory cytokines, also responsible for the neurogenic pruritus in dermatological conditions (sorry if I was repetitive)
Thank you for the pleasure of reading this paper.
Author Response
The authors have written a case report regarding the potential connection between neurological disease and myocardial lesion, and therefore, dysfunction.
Some minor comments below, and I will advise for publication.
Introduction
Line 50-51: first part of the phrase, before the comma, I think needs clarification with the “be”.
I see your point, so we rephrased that sentence as follow: “Neurological injuries in humans can cause myocardial alterations, which may manifest as reversible neurogenic left ventricular dysfunction, known as neurogenic stunned myocardium (NSM). Alternatively, these injuries can result in more severe conditions such as myocardial ischemia and infarction (MI).”
Case 1- Why did the authors decide to give AB, even though with normal WBC ?
Does the fever was related to the cardiac imbalance, or you believe could have had a bacterial involvement as well, since we had arterial punction?
The antibiotic therapy was started a few days prior to referral by the referring veterinarian due to the fever and lethargy of unknown origin. On admission, although no bacterial infections were suspected, the antibiotic treatment was continued to prevent secondary infections also because a pericardiocentesis was performed in an emergency basis.
Line 87: can you describe the technique for periocardiocentesis ? And in the text can you add a reference “as described..” ?
This has been done.
Case 2- One limitation is that no cardiac assessment has been made before, although, the clinical evidence is unquestionable (as above). Why the use of 7.5% hypertonic saline after the seizures? For potential brain edema?
Yes, that was the aim. We added in the explanation (line 201)
Line 170: there is a “8” after the final point.
It was a reference that lost the brackets. It has been corrected.
Line 192: "partial seizures”, can the authors please clarify and potentially point the difference between these episodes and the ones before?
Can the authors say what the window of access of video S3?
At the moment of the hypernatremia, was she still having seizures or neurological signs?
The first three episodes were generalized seizures with loss of consciousness and tonic-clonic contractions of the limbs (we added it at lines 191-192) while the successive episodes were only partial seizures with fasciculation and twitching of head and neck muscles together with spontaneous horizontal nystagmus (we added it at lines 202-204).
Supplementary item 3 is a right short-axis view at mitral level (added in the description at lines 439)
The hypernatremia developed after the resolution of all the neurological signs.
Line 216: there is “9” lost.
It was a reference that lost the brackets. It has been corrected.
Figure 3: no inflammatory infiltrate or cells were seen in the necrosis areas? I think the legend of figure 3 is not correct.
Can the authors edit table 1 and table 2 with same size and layout?
We thank the reviewer for their comment. The absence of neutrophils in the cardiac lesion is consistent with the acute nature of the myocardial injury in this case. According to the clinical history, the interval between the traumatic head injury and death was approximately 20 minutes. Given that neutrophil recruitment to sites of tissue injury generally begins after 30 minutes and tends to peak several hours later, the lack of an inflammatory infiltrate is in keeping with the very short time frame between the inciting traumatic event and death.
The legend has been changed, thank you.
The tables has been modified with the same layout and size.
I congratulate the authors for their work. The cases are very interesting and the pathological basis for the cardiac lesions is of deeper research ! One can speculate that the neurologic damage, affect the cardiac tissue. Further anatomopathological studies are needed in the site of the lesion, to verify the status of the innervation and cross results with the same analysis of the SNC area that send nerve branch’s to the heart. Potentially there is a neutrophilic inflammatory venue (giving the acute presentation). There is nerve fiber, SP positive, that are known to release inflammatory cytokines, also responsible for the neurogenic pruritus in dermatological conditions (sorry if I was repetitive)
Thank you for the pleasure of reading this paper.
Thank you for your kind words and very helpful review of the article.
Reviewer 2 Report
Comments and Suggestions for Authors
1. Need to add more references to support your research background
2. Can you add some additional cases?
Author Response
Need to add more references to support your research background
We added more references (one in the introduction and three in the discussion):
- Piliponis L, Neverauskaitė-Piliponienė G, Kazlauskaitė M, Kačnov P, Glaveckaitė S, Barysienė J, et al., editors. Neurogenic stress cardiomyopathy following aneurysmal subarachnoid haemorrhage: a literature review. Seminars in Cardiovascular Medicine; 2019.
- Ruiz Bailén M, Rucabado Aguilar L, López Martínez A. Aturdimiento miocárdico neurogénico. Medicina Intensiva. 2006;30(1):13-8. doi: https://doi.org/10.1016/S0210-5691(06)74456-4.
- Navas de Solis C, Dallap Schaer BL, Boston R, Slack J. Myocardial insult and arrhythmias after acute hemorrhage in horses. J Vet Emerg Crit Care (San Antonio). 2015;25(2):248-55. Epub 2015/03/11. doi: 10.1111/vec.12295. PubMed PMID: 25752472.
- Van Der Vekens N, Decloedt A, Ven S, De Clercq D, van Loon G. Cardiac troponin I as compared to troponin T for the detection of myocardial damage in horses. J Vet Intern Med. 2015;29(1):348-54. Epub 2015/01/27. doi: 10.1111/jvim.12530. PubMed PMID: 25619522; PubMed Central PMCID: PMCPMC4858065.
Can you add some additional cases?
That would have been awesome but we don’t have more cases well documented as the three presented.
Reviewer 3 Report
Comments and Suggestions for Authors
Dear authors,
After reading your case report entitle: "Can acute neurological disease cause cardiomyopathy in horses?"
There are few parts which could be improved. I understand that a detailed description of the administered drugs is required, however, it would be easier for the reader and for the veterinarians to have a shorten table with the main drug's used in both cases of the horses as a resume.
In addition to that, please find below some details that can improve your manuscript,
Best regards,
Reviewer suggestions:
L50-52: needs to be rephrase, not easy to understand, abbreviation of MI needs to be written after myocardial infarction (MI).
L60-62: could be improved, not that easy to follow when reading this paragraph
L147: please remove "with" between "presented" and "tachycardia"
L170: the number "8" between arrhythmias what does it stands for.
L216: the number "9" before "Hypertonic" what does it stands for, please remove it if not needed.
L255-259: this part seems to be evident due to post-mortem changes. Could you please indicate
the length of time of analysing after death.
L281-283: size letter seems to be higher in this part
L291: could you please indicate what you mean with "A cut off of 2.8ng/mL..." which cut-off are you referring, catecholamine? could you please re-write this part, it seems not that easy to follow up.
L295-296: please rephrase
Table2. Please add the abbreviation's meaning on the bottom of the table as in Table 1 and "*indicates altered result, should be placed on the bottom of the table
L379: please change 2dimensional to : Two-dimensional speckle tracking echocardiography (2DSTE) if that's the case
Conclusions: Verify letter size, seems smaller compared to discussion
Author Response
Dear authors,
After reading your case report entitle: "Can acute neurological disease cause cardiomyopathy in horses?"
There are few parts which could be improved. I understand that a detailed description of the administered drugs is required, however, it would be easier for the reader and for the veterinarians to have a shorten table with the main drug's used in both cases of the horses as a resume.
Thank you for your review. We addressed your comments and, if it’s ok for the editor we created a table with the drugs used in both cases.
In addition to that, please find below some details that can improve your manuscript,
Best regards,
Reviewer suggestions:
L50-52: needs to be rephrase, not easy to understand, abbreviation of MI needs to be written after myocardial infarction (MI).
We rephrased it as follow: “Neurological injuries in humans can cause myocardial alterations, which may manifest as reversible neurogenic left ventricular dysfunction, known as neurogenic stunned myocardium (NSM). Alternatively, these injuries can result in more severe conditions such as myocardial ischemia and infarction (MI).”
L60-62: could be improved, not that easy to follow when reading this paragraph
We improved the sentence as follow: “Cardiac troponin levels rise significantly more in myocardial infarction (MI) compared to neurogenic stunned myocardium (NSM). A cut-off value has been established to distinguish between these two conditions, which helps in assessing anesthetic risks and prognosis.”
L147: please remove "with" between "presented" and "tachycardia"
Done.
L170: the number "8" between arrhythmias what does it stands for.
It was a mistake. It has been removed.
L216: the number "9" before "Hypertonic" what does it stands for, please remove it if not needed.
It has been removed.
L255-259: this part seems to be evident due to post-mortem changes. Could you please indicate the length of time of analysing after death.
Thank you. This has been added.
L281-283: size letter seems to be higher in this part
Yes, it is, but this is the model provided by the journal, we did not change the size of the fonts.
L291: could you please indicate what you mean with "A cut off of 2.8ng/mL..." which cut-off are you referring, catecholamine? could you please re-write this part, it seems not that easy to follow up.
You are right. We were referring to cardiac troponine. We rephrased the sentence as follow: “A cut-off of 2.8 ng/mL for cardiac troponin has been established, indicating less risk of cardiovascular deterioration for patients with NSM”
L295-296: please rephrase
Yes, it has been rephrased as follow: “In a horse with experimentally induced MI, ST-segment changes were detected using a 12-lead ECG, resembling those observed in humans.”
Table2. Please add the abbreviation's meaning on the bottom of the table as in Table 1 and "*indicates altered result, should be placed on the bottom of the table
This has been done.
L379: please change 2dimensional to : Two-dimensional speckle tracking echocardiography (2DSTE) if that's the case
Yes, we changed it as you suggested.
Conclusions: Verify letter size, seems smaller compared to discussion
Yes, it is, but we just follow the model provided without changing letter sizes.